# A Novel Heterozygous Deletion Variant in *KLOTHO* Gene Leading to Haploinsufficiency and Impairment of Fibroblast Growth Factor 23 Signaling Pathway

**DOI:** 10.3390/jcm8040500

**Published:** 2019-04-12

**Authors:** Ernesto Martín-Núñez, Javier Donate-Correa, Caroline Kannengiesser, David-Paul De Brauwere, Christine Leroy, Claire Oudin, Gérard Friedlander, Carol Prieto-Morín, Víctor G. Tagua, Pablo A. Ureña-Torres, Juan F. Navarro-González

**Affiliations:** 1Unidad de Investigación, Hospital Universitario Nuestra Señora de Candelaria, Santa Cruz de Tenerife 38010, Spain; emarnu87@gmail.com (E.M.-N.); jdonate@ull.es (J.D.-C.); vtagua@funcanis.es (V.G.T.); 2Escuela de Doctorado y Estudios de Posgrado, Universidad de La Laguna, San Cristóbal de La Laguna 38200, Spain; 3GEENDIAB (Grupo Español para el estudio de la Nefropatía Diabética), Sociedad Española de Nefrología, Santander 39008, Spain; 4REDINREN (Red de Investigación Renal), Instituto de Salud Carlos III, Madrid 28029, Spain; 5Department of Genetics, CHU Paris Nord-Val de Seine, Hospital Xavier Bichat, Paris 75877, France; caroline.kannengiesser@aphp.fr (C.K.); claire.oudin@aphp.fr (C.O.); 6INSERM U1151-CNRS UMR8253, University Paris Descartes, and Functional Explorations Service, Hospital Necker-Enfants Malades, Paris 75015, France; David.Paul.De.Brauwere@chu-nimes.fr (D.-P.D.B.); christine.leroy@inserm.fr (C.L.); gerard.friedlander@inserm.fr (G.F.); 7Unidad de Genética Humana, Hospital Universitario Nuestra Señora de Candelaria, Santa Cruz de Tenerife 38010, Spain; cprimor@gobiernodecanarias.org; 8Nephrology-Dialysis Service, AURA Nord Saint Ouen, Saint-Ouen, 93400, France, and Department of Renal Physiology, Necker Hospital, University of Paris Descartes, Paris 75015, France; 9Servicio de Nefrología, Hospital Universitario Nuestra Señora de Candelaria, Santa Cruz de Tenerife 38010, Spain; 10Instituto de Tecnologías Biomédicas, Universidad de La Laguna, San Cristóbal de La Laguna 38200, Spain

**Keywords:** Klotho, deletion, haploinsufficiency, fibroblast growth factor 23, hyperphosphatemia, end-stage renal disease

## Abstract

Hyperphosphatemia is commonly present in end-stage renal disease. Klotho (KL) is implicated in phosphate homeostasis since it acts as obligate co-receptor for the fibroblast growth factor 23 (FGF23), a major phosphaturic hormone. We hypothesized that genetic variation in the KL gene might be associated with alterations in phosphate homeostasis resulting in hyperphosphatemia. We performed sequencing for determining *KL* gene variants in a group of resistant hyperphosphatemic dialysis patients. In a 67-year-old female, blood DNA sequencing revealed a heterozygous deletion of a T at position 1041 (c.1041delT) in exon 2. This variation caused a frameshift with substitution of isoleucine for phenylalanine and introduction of a premature termination codon (p.Ile348Phefs*28). cDNA sequencing showed absence of deletion-carrier transcripts in peripheral blood mononuclear cells suggesting degradation of these through a nonsense-mediated RNA decay pathway. Experiments in vitro showed that p.Ile348Phefs*28 variant impaired FGF23 signaling pathway, indicating a functional inactivation of the gene. In the patient, serum levels of KL were 2.9-fold lower than the mean level of a group of matched dialysis subjects, suggesting a compromise in the circulating protein concentration due to haploinsufficiency. These findings provide a new loss-of-function variant in the human *KL* gene, suggesting that genetic determinants might be associated to clinical resistant hyperphosphatemia.

## 1. Introduction

Hyperphosphatemia is a frequent condition in chronic kidney disease (CKD) patients, especially in subjects under dialysis therapy, and is recognized as a major cause of morbidity and mortality [1]. It can result from an increased phosphate intake, a shift of phosphate from the intracellular to the extracellular space, or more commonly due to a reduced renal phosphate excretion. Treatment is based on the reduction of phosphate intake with diet, the use of intestinal phosphate binders to reduce phosphate absorption, and the maximization of phosphate removal by dialysis. Nevertheless, hyperphosphatemia remains in some patients despite the optimization of these therapeutic strategies [2], leading to speculate that a genetic background might contribute to persistent hyperphosphatemia in some patients.

The fibroblast growth factor (FGF) 23/Klotho (KL) system has been recently characterized as a main regulator of mineral metabolism, with special relevance in phosphate homeostasis [3]. FGF23 is a secreted protein that promotes renal phosphate excretion and reduces serum active vitamin D levels. To exert its activity, FGF23 binds to FGF receptors, but needs the obligatory co-receptor KL, a protein mainly expressed in the kidneys that converts canonical FGF receptors into specific receptors for FGF23 [4].

KL was originally discovered in a mouse model where its locus was inactivated by insertional mutation of a transgene [5]. *Kl*-deficient mice showed a premature aging phenotype in addition to biochemical anomalies that resemble characteristics observed in patients with CKD, including severe hyperphosphatemia [6,7]. These similarities lead to speculate that genetic variants in the human *KL* gene could result in a disruption of renal phosphate excretion, and therefore contribute to the resistant hyperphosphatemia observed in some CKD patients despite the optimization of therapeutic strategies. To date, only one previous work has reported a homozygous missense disease-causing mutation in the *KL* gene (p.His193Arg (c.578A > G)) in a young female patient with severe hyperphosphatemic tumoral calcinosis [8]. Interestingly, a translocation reported to cause increased KL levels results in hypophosphatemic rickets and hyperparathyroidism [9].

To investigate this idea, we addressed the sequencing of the entire coding region of the *KL* gene in two groups of adult age and gender-matched dialysis patients that only differed in high or normal serum phosphate levels despite optimization of treatment. The present article reports, in a 67-year-old woman with hyperphosphatemia undergoing hemodialysis treatment, the identification of a novel heterozygous deletion variant in the *KL* gene (p.Ile348Phefs*28), possibly degraded by mRNA decay, that affects FGF23 signaling and leads to haploinsufficiency.

## 2. Materials and Methods

### 2.1. Ethics Statement and Human Samples

The study was approved by the Biomedical Research Ethic Committee of Bichat Claude Bernard Hospital (CRC 03161-P0331010) and conducted according to the Declaration of Helsinki principles. Written informed consent was obtained from the participants prior to inclusion in the study. Blood samples for the purposes of this study were collected before hemodialysis treatment.

### 2.2. Gene Sequencing and Variant Analysis

Genomic DNA was extracted from blood sample using the QIAamp DNA Blood Mini Kit (Qiagen, Hilden, Germany). All five coding exons, flanking intronic regions and part of the promoter region (−1500 bp to +1) of the KL gene (NM_004795.3) were amplified by PCR using a KAPA HiFi HotStart PCR Kit (Kapa Biosystems Inc., Boston, MA, USA). Each PCR product was purified using a QIAquick PCR purification kit (Qiagen) and subsequently sequenced from forward and reverse specific primers using a Big Dye Terminator Cycle Sequencing kit (Applied Biosystems, Foster City, CA, USA) and an ABI-PRISM 3500 Genetic Analyzer (Applied Biosystems). To study the sequence of KL gene transcripts, a blood sample was collected in a PAXgene Blood RNA tube (BD Diagnostics, Franklin Lakes, NJ, USA). Total RNA was isolated from peripheral blood mononuclear cells (PBMCs) using a PAXgene Blood RNA Kit (Qiagen) and stored at −80 °C. cDNA synthesis was performed using a High Capacity RNA-to-cDNA kit (Applied Biosystems). Sequencing of the transcripts was performed as previously explained with specific primers for RT-PCR products. CLUSTAL multiple sequence alignment was performed with MUSCLE (V.3.8) software [10] for the amino acidic sequences of *KL* wild-type and deletion variant. The potential pathogenic impact of the new allele identified on KL was investigated in silico with the MutationTaster software (http://www.mutationtaster.org/) [11].

### 2.3. Laboratory Measurements

Routine clinical biochemistry assays were measured by standard methods. Serum levels of KL protein were measured by a solid phase sandwich ELISA (Immuno-Biological Laboratories Ltd., Fujioka, Japan) according to manufacturer’s instructions. This assay detects the full-length extracellular domain (130 kDa) of the protein. The assay sensitivity was 6.15 pg/mL and the intra- and inter-assay coefficients of variation (CVs) were 3.1% and 6.9%, respectively. Serum levels of FGF23 were determined by using the second-generation C-terminal assay (Immutopics International, San Clemente, CA, USA), with a sensitivity of 1.5 relative units (RU)/mL and intra- and inter-assay CVs of 1.9% and 3.55%, respectively.

### 2.4. Expression Vector

The full-length and the p.Ile348Phefs*28 variant of human KL cDNAs were cloned into pcDNA6b expression vector (Invitrogen, Carlsbad, CA, USA). The p.Ile348Phefs*28 variant cDNA was generated using QuickChange® II Site-Directed Mutagenesis kit (Agilent Technologies, Santa Clara, CA, USA). All constructs were sequenced to verify introduction of the correct variants and the absence of cloning artifacts.

### 2.5. Cell Culture Experiments

HEK293 cell line used in this work was provided by American Type Culture Collection (ATCC). HEK293 cells were transfected with empty vector or containing full-length or p.Ile348Phefs*28 variant in the *KL* gene with Lipofectamine 3000 reagent (Invitrogen) and forced selected in culture medium DMEM/F12, bicarbonate 7.5%, gentamicin 10 mg/mL (GIBCO) and FBS 10% supplemented with blasticidine 5 μg/mL (AppliChem GmbH, Darmstadt, Germany). In order to evaluate activation of the FGF23 signaling pathway, HEK293 cells stably transfected with wild-type or mutant KL and maintained in selection media were seeded in 6-well plates (0.2 × 106 cells/well). Cells with empty vector served as negative control. All groups of cells (*n* = 3–8) were starved with serum-free medium plus 0.2% BSA for 16 h and then treated for 60 min with either recombinant FGF23 at 100 ng/mL (PeproTech Inc., London, UK) or vehicle (PBS with BSA 0.1%).

### 2.6. RNA Extraction and Quantitative PCR

Total RNA was extracted from the cells using RNAzol RT according to manufacturer’s instructions (Sigma Aldrich, St. Louis, MO, USA) and quantified using a Nanodrop Lite Spectrophotometer (Thermo Fisher Scientific, Waltham, MA, USA). 100 ng of RNA were reverse transcribed to cDNA using a High Capacity RNA-to-cDNA kit (Applied Biosystems) for further use in quantitative RT-PCR (qRT-PCR). Transcripts encoding for EGR1 and GAPDH were measured by TaqMan qRT-PCR with PerfeCTa FastMix II Low ROX (QuantaBio, Beverly, MA, USA). TaqMan gene expression assays for each transcript (Hs00152928_m1 (*EGR1*) and Hs99999905_m1 (*GAPDH*)) were analyzed in a 7500 Fast Real-Time PCR System (Applied Biosystems). The level of target mRNA was estimated by relative quantification using the 2^−ΔΔCt^ method and *GAPDH* as housekeeping gene. Quantification of each cDNA sample was tested in triplicate. A corresponding non-reverse transcriptase reaction was included as a control for DNA contamination.

### 2.7. Protein Extraction and Western Blot

Total protein from cell lysates were extracted with standard RIPA buffer (10 mM Tris-HCl pH 8.0, 140 mM NaCl, 1 mM EDTA, 1% Triton X-100, 0.1% sodium deoxycolate, 0.1% SDS) supplemented with PierceTM protease and phosphatase inhibitor mini tablets (Thermo Fisher Scientific) and further sonication with Ultrasonic-homogenizer LabsonicRM (Sartorius, Gotinga, Germany). The lysates were clarified by microcentrifugation at 10,000× *g* for 30 min at 4 °C, and protein concentration was quantified with Nanodrop Lite Spectrophotometer (Thermo Fisher Scientific) and the BCA assay (Abcam, Cambridge, UK). Aliquots of the cell lysates containing 30 μg protein were separated by SDS-PAGE and then Western blotted with antihuman α-extracellular signal-regulated kinases (ERK)1/2 monoclonal antibody (MA5-15134, Invitrogen) at a concentration of 1:1000, α-phospho-ERK1/2 (Thr202, Tyr204) polyclonal antibody (36–8800, Invitrogen) at a concentration of 1:250, and anti-β-actin (Cell Signaling Technology) at a concentration of 1:10,000. Light emission was detected with Clarify Western ECL Substrate (Bio-Rad, Hercules, CA, USA) on a Fusion Solo S chemiluminescence detector (Vilber Lourmat, Collégien, France).

### 2.8. Statistical Analysis

Continuous variables are reported as mean ± SD or median (interquartile range). Differences among groups were analyzed by one-way ANOVA test followed by Tukey’s multiple comparison test using GraphPad Prism 6.01 software (GraphPad Software, San Diego, CA, USA).

## 3. Results

### 3.1. Case Report

In a previous work, we addressed the influence of KL gene variants in recalcitrant hyperphosphatemia in two groups of adult age and gender-matched dialysis patients with high or normal serum phosphate levels despite optimization of treatment (Appendix A). A 67-year-old female patient originally from Guadeloupe (France), with CKD secondary to chronic interstitial nephropathy under maintenance hemodialyisis for 3 years, presented persistent hyperphosphatemia in spite of optimization of diet, treatment with oral phosphate binders and dialysis therapy (Table 1). Tubular reabsorption of phosphorous was inappropriately high (98% of filtered phosphate) in the setting of hyperphosphatemia.

### 3.2. Sequencing and Variant Analysis of KL Gene

DNA sequencing of the *KL* gene and part of its promoter region revealed that the patient in study had a heterozygous deletion of a thymidine (T) at position 1041 (c.1041delT) in exon 2 (Figure 1A). At protein level, this deletion disrupts the open reading frame with the subsequent substitution of an isoleucine by a phenylalanine at position 348. Furthermore, the frameshift caused by this deletion introduced a premature termination codon (PTC) 28 amino acids downstream (p.Ile348Phefs*28) (Figure 1B). The appearance of this PTC made us consider the possibility that the nonsense-mediated mRNA decay (NMD) pathway, a cellular surveillance mechanism that eliminates aberrant transcripts before they are translated into proteins, could act on the mRNAs of the c.1041delT variant. Sequencing of the cDNA from PBMCs of the patient did not show the presence of this deletion in *KL* transcripts suggesting that effectively the mRNA product of this allele is degraded by the NMD pathway (Figure 1C).

In order to study the pathogenic potential of the variant detected in the *KL* gene, we evaluated in silico its functional effect on the protein by using the MutationTaster software [11]. This program checks if the resulting protein of a particular genetic variant will be elongated, truncated, or whether nonsense-NMD is likely to occur. If the software concludes that a variant causes NMD, this alteration is automatically regarded as a disease mutation. In our new identified variant, the prediction resulted in a protein regulated by an NMD mechanism and, hence, to be "disease-causing" under a model of complex change of the amino acid sequence (i.e., a variant that introduces a PTC) with a probability of 1. Additionally, the software predicted the alteration of the amino acid sequence from residue 348 onwards and the appearance of the stop codon in residue 375. Consequently, the program pointed out that the main structural features of the protein were affected by the variant; this is that the functional regions KL1 and KL2 (residues 57–506 and 515–953, respectively), along with the transmembrane region (residues 982–1002) and the cytoplasmic domain (residues 1003–1012) are lost (Table 2). A proposed model explaining regulation of the *KL* gene expression with the deletion variant in the patient is shown in Figure 2.

### 3.3. Soluble KL and FGF23 Serum Levels

Soluble KL levels in our patient were diminished compared to adults with normal renal function (99.72 pg/mL vs. 562 ± 146 pg/mL) [12]. More interestingly, our patient presented a 2.9-fold lower concentration of serum KL compared to a group of 47 age and sex-matched hemodialysis subjects (99.72 pg/mL vs. 289.78 pg/mL). Regarding serum levels of C-terminal FGF23, the patient presented similar concentration to the mean levels in the matched dialysis group (1375.9 RU/mL vs. 1133.6 RU/mL). Finally, it is necessary to point that FGF23 was measured only by C-terminal ELISA, and that the determination of intact FGF23 would have provided significant complementary information since this form is a surrogate of functional levels of circulating FGF23.

### 3.4. FGF23 Signaling Pathway Activation

To assess whether the p.Ile348Phefs*28 variant in KL protein impairs the signaling pathway activated by FGF23, we stably transfected HEK293 cells with a pcDNA6b plasmid vector carrying either KL full-length, p.Ile348Phefs*28 variant or empty vector and studied if the molecular events in this pathway took place in the presence of FGF23 in the culture media. After a brief starvation (16 h) in serum-free medium plus 0.2% BSA, the cells were treated for 60 minutes with recombinant FGF23 (100 ng/mL) or vehicle. RNA and proteins were extracted for downstream analyses. Stimulation of cells with FGF23 induced phosphorylation of ERK1/2 in cells expressing wild-type *KL* while in those with mutated *KL* or empty vector there was no activation of these kinases (Figure 3A). Additionally, *early growth response* (*EGR*)*1* gene expression was 3.52-fold higher in cells with full-length *KL* treated with recombinant FGF23 when compared with treated cells carrying p.Ile348Phefs*28 variant (*p* < 0.001) and 3.51-fold higher respect to cells with empty vector (*p* < 0.01) (Figure 3B).

## 4. Discussion

Hyperphosphatemia is a harmful condition that predisposes to higher mortality in CKD patients [13]. Despite adequate dietary restrictions, treatment and compliance with phosphate binders, and optimal dialysis prescription, some patients show persistently elevated serum phosphate concentrations [2]. This permits the speculation that particular variants of genes essential for phosphate metabolism might be involved in the genesis or contribute to resistant hyperphosphatemia.

The FGF23/KL system is a fundamental axis of phosphate regulation. Human genetic variants in *FGF23* and *KL* loci have been reported to generate alterations in mineral metabolism [8,14,15,16,17,18,19,20]. To test the hypothesis that particular genetic variants of the *KL* gene could contribute to the existence of unexplained and resistant hyperphosphatemia in certain CKD patients, we addressed the sequencing of the coding region and promoter of *KL* in two groups of 20 dialyzed patients matched for age and gender, with comparable values of dialysis dose and daily protein intake, that were differentiated only regarding a normal (<1.60 mmol/L) or high (>1.60 mmol/L) serum phosphate levels. The serum levels of FGF23 and KL were similar between both groups.

In this study, we identified a previously unknown genetic variant in the *KL* gene in a 67-year-old hyperphosphatemic woman consisting of a heterozygous deletion of a T at position 1041 in exon 2 (c.1041delT). Such variant causes a shift in the open reading frame and generates a PTC downstream. At the protein level, the isoleucine at position 348 of the wild-type protein is substituted by a phenylalanine and the new stop codon establishes the early termination of the protein 28 amino acids downstream (p.Ile348Phefs*28). This variation is present at low frequency in control DNA databases (ExAC, MAF: 0.00001647).

The appearance of a PTC in exon 2 makes plausible the idea that the putative transcript carrying the deletion allele is susceptible to be degraded by the NMD pathway. NMD is a cellular surveillance system which efficiently removes mRNAs with PTCs that otherwise would lead to the production of truncated proteins with potential deleterious effects [21]. In mammals, this system is based in a proper pre-mRNA splicing where exon junction complexes (EJCs) are involved. These complexes are deposited 20–24 nucleotides upstream of exon-exon junctions after RNA splicing and they remain associated until ribosome displaces them during the first round of translation. If any EJC remains attached to the RNA after the pioneer round of translation (e.g., due to a PTC upstream of the last exon-exon junction that would trigger early dissociation of the ribosome), this acts as a signal to elicit mRNA wipe out [21,22]. To study the possibility that NMD pathway acts upon the *KL* mRNAs harboring the deletion, we proceeded to sequence *KL* gene transcripts present in PBMCs from the patient. Sequencing of the cDNA obtained from these cells revealed the exclusive presence of transcripts with the wild-type allele, supporting the idea of degradation of deletion-carrier mRNAs. Another interesting approach would have been to study if the putative truncated form of KL protein was produced by the patient and if this coexisted with wild-type forms. Unfortunately, by the time we were developing the experimental work of this study, the patient had already died. Therefore, it was not possible to obtain new samples (this approach was not originally conceived for the purposes of the study). This fact is a limitation regarding the conclusion about whether the deletion variant was translated or not.

Additionally, we further evaluated the potential effects of this new genetic variant in silico by testing it with MutationTaster, a software designed to predict functional consequences of amino acid substitutions, intronic and synonymous alterations, short indels and variants spanning intron-exon borders [11]. Interestingly, the software cataloged the p.Ile348Phefs*28 variant to be “disease-causing” since the protein is likely to be regulated by a NMD mechanism, as we previously hypothesized. Likewise, the program points out that an alteration of the amino acid sequence of the new variant protein occurs from residue 348 and up to residue 375, where interruption of the protein takes place due to a stop codon. The full-length KL protein is a 1012 amino acids polypeptide with a molecular weight of 130 kDa, and is constituted by a large extracellular domain that covers most of the chain (residues 34–981), a transmembrane helical region (residues 982–1002) and a short cytoplasmic domain (residues 1003–1012). In the extracellular domain, the protein includes two glycosyl hydrolase regions designated KL1 (residues 57–506) and KL2 (residues 515–953) that may have weak glycosidase activity. So, as the software considered regarding the alteration in the amino acid sequence, in case that this new variant protein was synthesized, a truncated KL protein of 375 amino acids and 43 kDa would be generated that would lose part of the extracellular domain (interrupted towards the middle of the KL1 region and lacking the entire KL2 region), the single membrane-spanning region and the cytosolic domain, completely.

This loss of so many domains in the putative protein variant would have important consequences on its performance. Among its described functions, KL has an important role in the maintenance of phosphorus homeostasis due to its participation in signaling pathways activated by FGF23, a major phosphatonin [23,24]. The presence of KL in a tissue is mandatory so that it is sensitive to the effects of FGF23. Actually, KL null mice develop all the phenotypes observed in Fgf23^−/−^ mice [5]. This is explained by the fact that KL is a co-receptor of the FGF receptors [25]. When FGF23 binds to the FGFR1c-KL membrane complex, it triggers the intracellular mitogen-activated protein kinase (MAPK)/ERK signaling pathway which involves ERK1/2 phosphorylation and, eventually, upregulation of transcription regulators like the *EGR1* gene [23,25]. The crystal structure of human 1:1:1 FGF23–FGFR1c^ecto^–α-KL^ecto^ ternary complex at 3.0 Å resolution has been recently described [26]. According to this, both KL regions serve as a massive scaffold, tethering both FGFR1c and FGF23 to itself, enforcing FGF23-FGFR1c proximity and thus augmenting FGF23-FGFR1c binding affinity. Several residues in KL protein were described as essential for FGF23 signaling and most of them are missed in the putative p.Ile348Phefs*28 protein, like the receptor binding arm (RBA, residues 530–578); amino acids D426, D745, C739 and D815 that bound Zn^2+^ ion serving as a prosthetic group in KL by minimizing interdomain flexibility and hence promoting co-receptor activity; and residues W417, K429, Y433, D756, I822 and I836 that participate in tethering of the flexible C-terminal tail of FGF23. Without all these residues, FGF23 signaling would be not possible since the interaction between FGF23, FGFR1c and KL would not take place.

To consider this loss of function of KL due to the new deletion variant, we also interrogated the impairment of FGF23-activated signaling in an in vitro model of HEK293 cell line transfected with either the KL full-length or the p.Ile348Phefs*28 variant. The human HEK293 cell line is particularly relevant for this experiment since endogenously it does express FGFR1c, the main receptor of FGF23, but do not express the Klotho protein [27,28]. Our experimental approach showed that with recombinant FGF23 present in the cell media, phosphorylation of ERK1/2 only took place in HEK293 cells carrying plasmids with the KL full-length form, whereas this did not happen in either cells with p.Ile348Phefs*28 plasmid or with empty vector. Moreover, upregulation of the *EGR1* gene due to recombinant FGF23 was only observed in cells harboring full-length KL plasmid. Altogether, these evidences point out that p.Ile348Phefs*28 variant leads to inactivation of *KL* gene and, subsequently, to impairment of FGF23 signaling pathway. Although these findings are according to our hypothesis, we recognize as a limitation that the study of the FGF23-induced signaling pathways would be ideally analyzed in at least two different cell lines. However, to the best of our knowledge, there are no other human cell lines that share these characteristics with HEK293 cells: absence of Klotho expression and presence of FGFR1c.

In addition to its location in the cellular surface as a transmembrane protein, KL can be found in a soluble form detectable in serum, urine and cerebrospinal fluid [29]. This soluble form can be generated either from the proteolytic cleavage of the transmembrane form by membrane secretases, so that the full-length extracellular domain is released into the circulation, or from direct secretion of an alternative splicing variant product [30,31]. The main producer of soluble Klotho in the organism is the kidney [4] and the role of this protein as an early and sensitive biomarker of decline in renal function has been suggested [32]. Since our patient was enrolled in hemodialysis treatment, as expected, her soluble serum KL levels were importantly diminished (5.64-fold lower) compared to average adults with normal renal function [12]. However, and interestingly, when compared with a group of 47 age and sex-matched hemodialysis subjects, our patient also presented an evident reduction in serum KL (2.9-fold lower concentration). Although there are no studies that analyze the relationship between the Klotho gene dosage and tissue expression and circulating levels of soluble KL, the values found in our patient might suggest that serum concentrations, as well as renal KL expression are compromised since there is only one wild-type *KL* allele available to produce the full-length protein. In spite of the severely decreased renal function in our patient, the reduction of functional KL with the consequent insensitivity to the action of FGF23 could partially explain the state of persistent and resistant hyperphosphatemia described in the patient.

## 5. Conclusions

Here we report a novel heterozygous deletion variant in the *KL* gene in a hemodialysis patient with persistent hyperphosphatemia that leads to inactivation of the gene since its transcript is presumably degraded through NMD pathway due to a PTC introduced in exon 2. To the best of our knowledge, this is the second loss-of-function genetic variant in the human *KL* gene to be described and, in addition, to produce alterations in mineral metabolism. Since Klotho deficiency is a condition associated with different pathological scenarios, such as CKD or cardiovascular disease [6,7,33], it would be interesting to perform studies that investigate the possible association of this and other non-functional variants of the *KL* gene with these diseases. Furthermore, it does not escape to us that the absence of certain experiments is a limitation to this work since it would have allowed to validate our results. For example, it would have been desirable to perform Klotho protein expression studies in additional samples from the patient to confirm the absence of a putative truncated protein, to have validated the degradation of mRNA in renal biopsies (since the kidney is the principal organ expressing KL), and to have analyzed FGF23-activated signaling pathway in other cell lines. Nonetheless, our study supports the idea that exploration of possible genetic causes should be addressed to explain cases of resistant hyperphosphatemia despite the optimal adjustment of therapeutic interventions in dialysis patients.

## Figures and Tables

**Figure 1 jcm-08-00500-f001:**
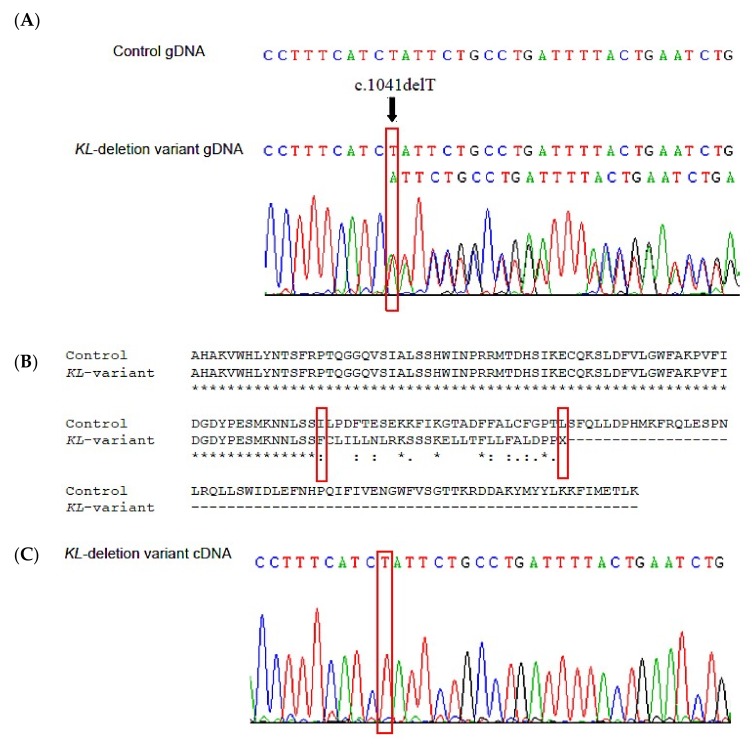
Analysis of deletion variant in the *KL* gene. (**A**) DNA sequencing of the *KL* gene revealed deletion of a T at 1041 in exon 2. Up, wild-type control; down, patient with heterozygous deletion. (**B**) CLUSTAL multiple sequence alignment by MUSCLE (Ver. 3.8) of the predicted amino acid sequences of control and patient deletion allele showing the substitution of isoleucine-348 for phenylalanine, the reading frame shift and the appearing of a premature stop codon. (**C**) cDNA sequencing of the region corresponding to exon 2 in PBMCs of the patient showing the permanence of the T.

**Figure 2 jcm-08-00500-f002:**
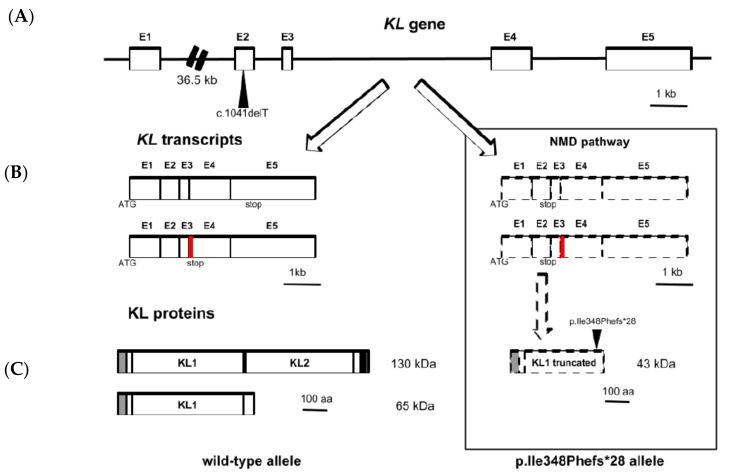
*KL* gene expression with the heterozygous p.Ile348Phefs*28 variant. (**A**) *KL* gene (50 Kb) is structured in five exons with c.1041delT allele located within exon 2. (**B**) *KL* processed transcripts carrying wild-type allele (left) can be found as complete mRNAs (coding for exons 1–5) or as shorter splicing variants (coding for exons 1-4, with an alternative sequence between exons 3–4 indicated in red). *KL* transcripts carrying c.1041delT allele (right) are degraded through an NMD pathway due to the introduction of a premature termination codon in exon 2 (dashed line). (**C**) The transcripts are translated into a full-length transmembrane KL protein (130 KDa) with two functional domains (KL1 and KL2) that can generate a soluble form from its shedding by membrane metalloproteases or into a shorter soluble form (only with KL1 domain), encoded by the alternative splicing transcript, which is directly secreted from the cell (65 KDa) (left). The putative KL protein with the p.Ile348Phefs*28 variant would be a shorter form (43 KDa) with a truncated KL1 domain (right).

**Figure 3 jcm-08-00500-f003:**
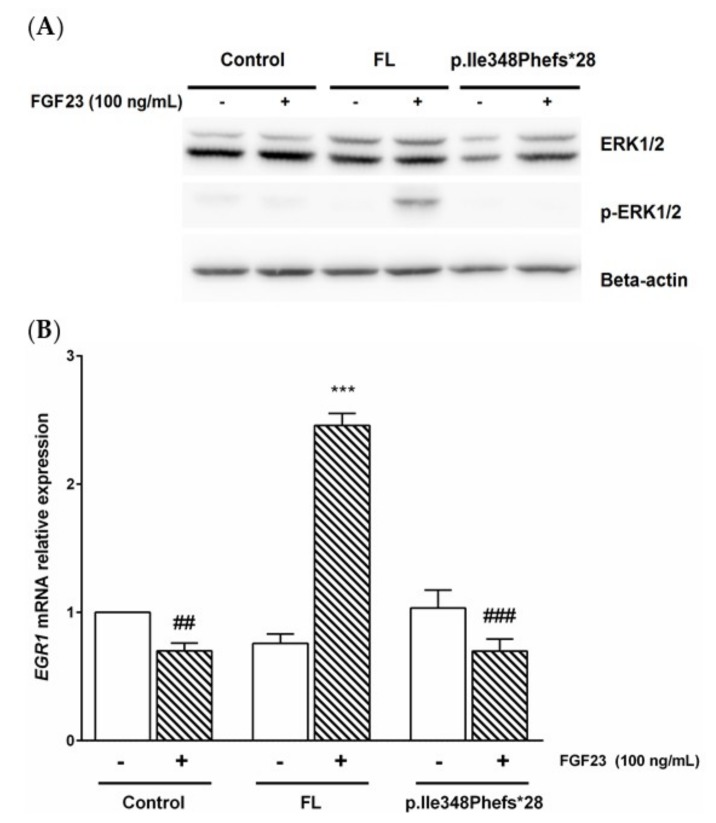
The p.Ile348Phefs*28 variant in *KL* gene impairs FGF23 signaling pathway. HEK293 cells were transfected with empty vector (Control), full-length (FL) or p.Ile348Phefs*28 *KL* gene variant and treated with vehicle (PBS with BSA 0.1%; −) or recombinant FGF23 (100 ng/mL; +) for 60 min. (**A**) Immunoblot of total and phosphorylated ERK1/2 (*n* = 3 for each condition). Molecular weights for ERK1, ERK2 and β-actin are, respectively, 44 kDa, 42 kDa and 45 kDa. (**B**) Relative *EGR1* gene expression assessed by quantitative RT-PCR using *GAPDH* as the housekeeping gene (*n* = 8 for each condition). Results are presented as fold change (mean ± SEM). *** *p* < 0.001 vs. vehicle, ^##^
*p* < 0.01 or ^###^
*p* < 0.001 vs. *KL*-FL+FGF23.

**Table 1 jcm-08-00500-t001:** Laboratory data of the patient with heterozygous deletion variant in the *KL* gene.

	Normal Ranges	Patient Values
Calcium (mM)	2.25–2.60	2.5
Albumin (g/L)	30–42	35.6
Phosphorous (mM)	0.85–1.5	2.7
Potassium (mM)	3.5–4.5	5.8
Sodium (mM)	136–146	135
Bicarbonate (mM)	22–29	21
Chloride (mM)	98–106	101
Glucose (mM)	3.9–5.8	4.8
Urea (mM)	2.5–7.5	27.6
Creatinine (µM)	62–106	620
eGFR (mL/min)	>60	<5
PTH (pg/mL)	10–46	533
25(OH)D_3_ (ng/mL)	30–80	21.5
1,25(OH)_2_D_3_ (pg/mL)	15–60	30
Alkaline phosphatase (IU/L)	30–160	25.1
C-terminal Telopeptide (pg/mL)	650–5300	2580
C-terminal FGF23 (RU/mL)	1–149	1375.9
KL (pg/mL)	239–1266	99.72

eGFR, estimated glomerular filtration rate; FGF23, fibroblast growth factor 23; PTH, parathyroid hormone; KL, Klotho.

**Table 2 jcm-08-00500-t002:** Summary of outcomes from the in silico analysis of the impact of c.1041delT variant in KL protein in the Mutation Taster software.

Issue	Result				
Length of Protein	NMD				
AA Sequence Altered	Yes				
AA Changes	I348Ffs*28				
Position of Altered AA	348 (frameshift or PTC)				
Position (AA) of Stopcodon in Wild-Type/Mutated AA Sequence	1013/375				
Protein Features Affected	Start (AA)	End (AA)	Feature	Details	Result
	34	981	TOPO_DOM	Extracellular	Lost
	57	506	REGION	Glycosyl hydrolase-1.1.	Lost
	497	497	MOD_RES	Phosphotyrosine	Lost
	515	953	REGION	Glycosyl hydrolase-1.2.	Lost
	607	607	CARBOHYD	N-linked (GlcNAc)	Lost
	612	612	CARBOHYD	N-linked (GlcNAc)	Lost
	694	694	CARBOHYD	N-linked (GlcNAc)	Lost
	982	1002	TRANSMEM	Helical	Lost
	1003	1012	TOPO_DOM	Cytoplasmic	Lost

AA: amino acid; NMD: nonsense-mediated mRNA decay; PTC: premature termination codon; TOPO_DOM: topological domain; MOD_RES: modified residue; CARBOHYD: carbohydrate attached group; TRANSMEM: transmembrane.

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
