# Peer review of "A Novel Heterozygous Deletion Variant in KLOTHO Gene Leading to Haploinsufficiency and Impairment of Fibroblast Growth Factor 23 Signaling Pathway"

_jcm, 2019, doi:10.3390/jcm8040500_

Reviewer 1 Report

Manuscript reports a novel heterozygous deletion variant in Klotho gene in a hemodialysis patient with persistent hyperphosphatemia. While the described loss-of-function genetic variant of human KL gene is novel and interesting, provided data are not solid enough to implicate the high clinical importance of identified mutation.

Several key experiments are missing, including:

- further experiments in selected tissue confirming the hypothesis that the transcript of mutated allele is degraded by the NMD pathway;

- proteomic study of selected patient tissues (with expected high level of KL protein);

- extended in silico analysis of the impact of identified mutation;

- transgenic (mouse) model for studying in vivo 1041delT mutation;

- more detailed analysis of FGF23-induced signaling pathways in at least two different cell lines.

In my opinion the manuscript is not well suited for JCM journal.

Author Response

We appreciate the reviewer’s comments and the observation about the novelty and interest of the variant of the KLOTHO gene (KL) that we describe in this article. We absolutely agree that more experiments would have been necessary. However, unfortunately, at the time of describing this novel genetic variant the patient had already died, and therefore we couldn’t do additional analysis.

Regarding the experiments suggested by the reviewer, we have some comments:

 1) Experiments in selected tissue confirming the hypothesis that the transcript of mutated allele is degraded by the NMD pathway, and 2) proteomic study of selected patients’ tissues (with expected high level of KL protein).

 We agree that this approach is key to validate the molecular mechanism proposed for the regulation of the identified gene variant (degradation of mRNA by the NDM pathway and absence of a putative protein), especially in renal tissue samples since the kidney is the principal organ expressing KL, and main source of soluble KL in the body. However, at the time of developing the experimental work of the study the patient had already died, and therefore we were unable to obtain new biological material. This has been specified in the discussion section of the article and it is recognized as a limitation (page 8, lines 271-274).

 3) Extended in silico analysis of the impact of identified mutation.

 Since the identified variant is a missense type with the appearance of a premature stop codon (PTC), it is not expected that an extended in silico analysis provides relevant information beyond the degradation of the mRNA through the NMD pathway. This would be definitely interesting in the case of missense mutations in which the protein is produced. Nevertheless, we decided to perform an in silico approach in order to validate the aforementioned prediction.

 4) Transgenic (mouse) model for studying in vivo 1041delT mutation.

 Obtaining a transgenic mouse model carrying the 1041delT variant would be a tool of high interest to understand the molecular mechanisms and the physiological impact of the mutation. However, this is a complex project both from a biological and logistic (including economic) perspective that we can’t address at the present time, but it has become one of our main aims for the near future.

On the other hand, we believe that this approach is beyond the descriptive objective at the clinical and molecular level that the present study intends to cover.

 5) More detailed analysis of FGF23-induced signaling pathways in at least two different cell lines.

Undoubtedly, this approach in different cell lines, including renal cells, would be of interest to verify the impairment on the FGF23-signaling pathways due to inactivation of the KL gene by the identified variant. Unfortunately, as previously commented, we do not have additional biological material in our laboratory to reach that point.

 Considering the comments and suggestions by the reviewer, we have included in the discussion of the article a mention about the limitation that the absence of these experiments implies in our study (page 9, lines 323-332). Despite this, we consider that the data presented here support the idea that the genetic background could be involved in certain cases of persistent hyperphosphatemia in end-stage renal disease patients. Further studies would be necessary to delve into this issue. In any case, we would like to highlight the potential clinical relevance of this KL gene variant in this scenario. This is the second loss-of-function genetic variant in human KL gene described to date. The first one was described in a 13-year-old girl presenting hyperphosphatemic familial tumoral calcinosis, a rare autosomal recessive metabolic disorder. However, in our study, the patient was a 67-year-old woman with chronic kidney disease, a global health burden with a prevalence higher than 10%, in which hyperphosphatemia is a common complication with a reported prevalence in dialysis patients between 50% and 70%. In our study, we found this genetic variant in KL gene after studying a small group of dialysis patients with resistant hyperphosphatemia. Considering that over 2 million people worldwide receive renal replacement therapy for end-stage renal disease, overall these data suggest that this novel genetic variant could be more frequent than we think.

 Finally, we disagree with the reviewer regarding his opinion about the suitability of our manuscript for JCM. We consider that there are several reasons to consider that our manuscript is suited for JCM: 1) The aim of JCM is provide a platform for new advances in Medicine based on the study of direct observation of patients and general medical research. Our study comes from the study of patients with a high prevalent disease and a frequent complication, and it represents an advance in the knowledge since it describes a novel genetic variant causing a loss-of-function in human KL gene. In addition, it is an example of the interplay between genetics and clinics; 2) This manuscript fits into the Nephrology and Urology section of JCM; 3) As the journal encourage, and taking into account the limitations derived from the inability to obtain new samples from the patient, our study presents the results in as much detail as possible, with the use of a number of techniques: gene sequencing, enzyme-linked immunosorbent assays, mutagenesis, cell cultures, in silico analysis, etc.; 4) Our study has potential clinical translation, since this novel genetic variation represents a new diagnostic possibility for patients with resistant hyperphosphatemia; 5) The results presented in this manuscript represent the basis for new studies, both from experimental and clinical perspectives; and 6) our manuscript is of potential interest for multi-disciplinary professionals, including nephrologists, geneticists, endocrinologists, oncologists, etc.

Reviewer 2 Report

Martin-Nunez and colleagues report their manuscript “A novel heterozygous deletion variant in Klotho gene leading to haploinsufficiency and impairment in FGF23 signaling pathway”. I congratulate the authors for this well-presented work and contribution. Here are my comments to improve their report: 1- The authors identified a new loss-of-function variant in the human Klotho gene and proposed the occurrence of a truncated KL protein (KL1, 43 kDa) through the NMD pathway. Can this aberrant protein be an alternative soluble Klotho variant that can further risk-stratify patients (e.g., refractory hyperphosphatemia, CKD/CVD progression, risk of AKI? The authors should further comment about potential applications of their findings in additional research and clinical practice 2- As biospecimens from the reported patient are no longer available, did the authors attempt to measure this new variant in serum from a similar phenotype of ESRD patients with refractory hyperphosphatemia? 3- Did the authors consider other in vitro experiments to test other actions of soluble Klotho (e.g., anti-fibrosis, suppression of apoptosis, upregulation of autophagy, etc.? 4- Do the authors have data on serum intact FGF23 levels? This may enhance their report being intact FGF23 a surrogate of functional levels of circulating FGF23 5- When reporting data on the ESRD patient with the identified mutation, the authors should specify that the biochemical parameters were measured before hemodialysis (if so) 6- Please clarify if the “n” reported in the legend of Figure 3 is for each group of conditions 7- It will be informative if the authors summarize data of comparisons between the matched ESRD patients with different levels of serum phosphate, particularly for soluble Klotho and FGF23 levels, and other related biochemical parameters 8- In addition to transfecting cultured cells with a plasmid vector carrying the new Klotho variant, did the authors consider doing similar in vitro experiments measuring ERK1/2 phosphorylation and/or EGR1 gene expression by exposure to full length Klotho protein (KL1+KL2), functional KL1 (65 kDa), truncated/aberrant KL1 fragment (43 kDa) and control

Author Response

We appreciate the kind comments and suggestions of the reviewer. We have the following responses:

 1-The authors identified a new loss-of-function variant in the human Klotho gene and proposed the occurrence of a truncated KL protein (KL1, 43 kDa) through the NMD pathway. Can this aberrant protein be an alternative soluble Klotho variant that can further risk-stratify patients (e.g., refractory hyperphosphatemia, CKD/CVD progression, risk of AKI? The authors should further comment about potential applications of their findings in additional research and clinical practice.

 Actually, we propose that the putative aberrant protein does not get generated since the NDM pathway acts at the mRNA level degrading the transcripts that harbor the 1041delT variant. This idea is confirmed by sequencing of mRNA transcripts in PBMCs that show the exclusive presence of the wild type transcript in these cells (and not the new identified variant). A limitation that we point out in the discussion section (page 8, lines 271-274) is that we could not access to new samples from the patient in order to study the protein expression of the different Klotho isoforms and, thus, be able to definitively confirm or deny the non-production of the aberrant protein. Therefore, it is not possible to measure the levels of this putative variant of the KL protein and, consequently, to study its potential use as a marker for risk-stratification of the patients. However, it would be interesting to evaluate in future studies the association of this non-functional variant of the KL gene with the prevalence and/or evolution of diseases related to alterations in Klotho (CVD, AKI, CKD). We have included this observation in the conclusions of the article (page 9, lines 323-326).

 2-As biospecimens from the reported patient are no longer available, did the authors attempt to measure this new variant in serum from a similar phenotype of ESRD patients with refractory hyperphosphatemia?

 We appreciate this question, and the answer is “yes”. In fact, we are now working in the design of a multicenter study to analyze this novel variant in the KL gene in a large group of patients with end-stage renal disease and resistant hyperphosphatemia. The finding of this genetic variant in other patients would allow us to perform all the additional experiments that could not be done due to the lack of biological samples from the initial patient.

 3-Did the authors consider other in vitro experiments to test other actions of soluble Klotho (e.g., anti-fibrosis, suppression of apoptosis, upregulation of autophagy, etc.?)

 Our research interest regarding Klotho has been focused on the role of this protein in atherosclerotic vascular disease, initially in subjects with normal renal function. We have described the expression of KL in the human vascular wall (Donate-Correa et al. Int J Cardiol 2013), we have described that reduced Klotho, both serum concentrations and vascular expression levels, is independently associated with the presence and severity of coronary artery disease (Navarro-González et al. Heart 2014), we have reported that KL gene polymorphisms may be related to vascular Klotho expression and cardiovascular disease (Donate-Correa et al. J Cell Mol Med 2016), and more recently, that the soluble levels and the endogenous vascular gene expression of KL are related to inflammation in human atherosclerosis (Martín-Núñez et al. Clin Sci 2017). All these findings have led us to develop an experimental model of vascular injury in renal disease (Cazaña-Pérez et al. Front Physiol 2018) with the aim to characterize the physiopathological implication of Klotho and the actions of the soluble form of this protein. We have also considered to address these aspects in the renal tissue. We are very grateful for the very interesting suggestions by the reviewer.

 4-Do the authors have data on serum intact FGF23 levels? This may enhance their report being intact FGF23 a surrogate of functional levels of circulating FGF23

 Specific recognition of the intact form of FGF23 protein (iFGF23) is achieved by using the combination of two antibodies that simultaneously recognize the N- and the C-terminal domains of the protein. However, the C-terminal ELISA assay recognizes both iFGF23 as well as C-terminal fragments derived from FGF23 proteolysis. In this study, we only employed a C-terminal ELISA assay for determining FGF23 levels. Unfortunately, we don’t have any other serum sample from the patient to measure iFGF23. Undoubtedly, this point is very interesting and it would have provided significant complementary information. This aspects is now commented in the manuscript (page 6, line 210; page 7, lines 211-212).

 5-When reporting data on the ESRD patient with the identified mutation, the authors should specify that the biochemical parameters were measured before hemodialysis (if so)

 In the “Material and methods” section, subsection “2.1. Ethics statement and human samples” we have now specified that the samples were obtained before hemodialysis:  ”Blood samples for the purposes of this study were collected before hemodialysis treatment” (page 2, lines 84-85).

 6-Please clarify if the “n” reported in the legend of Figure 3 is for each group of conditions

 We have clarified this point in the legend of figure 3.

 7-It will be informative if the authors summarize data of comparisons between the matched ESRD patients with different levels of serum phosphate, particularly for soluble Klotho and FGF23 levels, and other related biochemical parameters.

 Comparison of mineral metabolism parameters between the two groups of hemodialysis patients matched by age and sex but with different serum phosphate levels showed that the only different data besides serum phosphate was the serum intact PTH concentration, which was, as expected, higher in the group of subjects with hyperphosphatemia. However, there were no differences regarding FGF23 and Klotho levels. This aspect has been specified in the manuscript (page 8, line 248). In addition, a table showing the comparison of mineral metabolism parameters between both groups has been included as supplementary material (Table S1).

8-In addition to transfecting cultured cells with a plasmid vector carrying the new Klotho variant, did the authors consider doing similar in vitro experiments measuring ERK1/2 phosphorylation and/or EGR1 gene expression by exposure to full length Klotho protein (KL1+KL2), functional KL1 (65 kDa), truncated/aberrant KL1 fragment (43 kDa) and control

 Transmembrane Klotho acts as co-receptor for FGF23, and therefore it participates in the FGF23 downstream signaling pathway, which includes ERK1/2. However, as far as we know, there are no studies in which the signaling pathway ERK1/2 is associated with soluble Klotho. The suggestion by the reviewer is very interesting and we are considering including this assay in an upcoming study by testing the full length and the functional KL1 proteins.

However, in this work we argue that the heterozygous deletion of a T at position 1041 in exon 2 (c.1041delT) describes in this study causes a shift in the open reading frame and generates a PTC downstream. This determines that the putative transcript carrying the deletion allele is degraded by the NMD pathway, which prevents the production of the truncated protein with potential deleterious effects. This is supported by the analysis of the sequences of the KL gene transcripts present in peripheral blood mononuclear cells from the patient, which revealed the exclusive presence of transcripts with the wild-type allele, supporting the idea of degradation of deletion-carrier mRNAs.

Despite this consideration, the reviewer’s comment is important, and would have been interesting to study if the putative truncated form of KL protein was produced by the patient and if this coexisted with wild-type forms. This approach was not originally conceived for the purposes of the study and we couldn’t get new samples since the patient had passed away. However, we further evaluated the potential effects of this new genetic variant in silico by testing the implications of the deletion with MutationTaster. This software predicted the loss of multiple regions of the protein: functional domains KL1 and KL2, the transmembrane region and the cytoplasmic domain.

 All these aspects are already included in the discussion section of the manuscript.

Round  2

Reviewer 1 Report

The revised version of the manuscript does not include any additional experiments and none of the points I raised have been taken into account by the authors.

The authors claim that they cannot provide more detailed analysis because "the patient had already died", and therefore they were "unable to obtain new biological material". Such explanation does not justify the lack of solid data. Further, in the authors' opinion in silico analysis of the impact of identified mutation (premature stop codon) would not provide any relevant information. My comment concerned the bioinformatics prediction of the structure and confirmation of truncated protein and the loss of their interaction sites. Such analysis would be valuable, especially in the absence of experimental results.

I am also disappointed with the authors' response to my request about detailed analysis of FGF23-induced signaling pathways in at least two different cell lines. The authors claim that they cannot perform additional experiments because they have no longer biological material at their disposal. This is completely unrelated, because the experiments I asked for are in vitro experiments in transfected cell lines (widely available). 

 Minor issue:

The intensity of phospho-ERKs WB is very weak. If n=3, more exposed experiment should be presented.

Author Response

First of all, we would like to express our apologizes to the reviewer because we had not understood his comments completely.

 1) Bioinformatic analysis.

According to the reviewer’s suggestion, we have performed a more detailed bioinformatic study that includes the prediction of the structure of the putative variant protein and confirmation of the loss of its domains. According to this re-analysis we have modified the Results section (changes are in lines 173-176 and 188-200). In addition, we have included a new table that summarizes the main outcomes in the prediction of the protein structure (Table 2):

 Table 2. Summary of outcomes from the in silico analysis of the impact of c.1041delT variant in KL protein in the Mutation Taster software.

Issue

Result

Length   of protein

NMD

AA sequence altered

Yes

AA changes

I348Ffs*28

Position of altered   AA

348 (frameshift or   PTC)

Position (AA) of   stopcodon in wild-type / mutated AA sequence

1013 / 375

Protein features   affected

Start (AA)

End (AA)

Feature

Details

Result

34

981

TOPO_DOM

Extracellular

Lost

57

506

REGION

Glycosyl hydrolase-1.1.

Lost

497

497

MOD_RES

Phosphotyrosine

Lost

515

953

REGION

Glycosyl   hydrolase-1.2.

Lost

607

607

CARBOHYD

N-linked (GlcNAc)

Lost

612

612

CARBOHYD

N-linked (GlcNAc)

Lost

694

694

CARBOHYD

N-linked (GlcNAc)

Lost

982

1002

TRANSMEM

Helical

Lost

1003

1012

TOPO_DOM

Cytoplasmic

Lost

AA: aminoacid; NMD: nonsense-mediated mRNA decay; PTC: premature termination codon; TOPO_DOM: topological domain; MOD_RES: modified residue; CARBOHYD: carbohydrate attached group; TRANSMEM: transmembrane.

 Finally, we have also modified the Discussion section in order to detail the implications derived from the synthesis of this mutated protein (lines 300-315 and 322-332), and we have added a new reference (#26). The modified paragraph is as follow:

 "Additionally, we further evaluated the potential effects of this new genetic variant in silico by testing it with MutationTaster, a software designed to predict functional consequences of amino acid substitutions, intronic and synonymous alterations, short indels and variants spanning intron-exon borders [22]. Interestingly, the software cataloged the p.Ile348Phefs*28 variant to be "disease-causing" since the protein is likely to be regulated by a NMD mechanism, as we previously hypothesized. Likewise, the program points out that an alteration of the amino acid sequence of the new variant protein occurs from residue 348 and up to residue 375, where interruption of the protein takes place due to a stop codon. The full-length KL protein is a 1012 amino acids polypeptide with a molecular weight of 130 kDa, and is constituted by a large extracellular domain that covers most of the chain (residues 34-981), a transmembrane helical region (residues 982-1002) and a short cytoplasmic domain (residues 1003-1012). In the extracellular domain, the protein includes two glycosyl hydrolase regions designated KL1 (residues 57-506) and KL2 (residues 515-953) that may have weak glycosidase activity. So, as the software considered regarding the alteration in the amino acid sequence, in case that this new variant protein was synthesized, a truncated KL protein of 375 amino acids and 43 kDa would be generated that would lose part of the extracellular domain (interrupted towards the middle of the KL1 region and lacking the entire KL2 region), the single membrane-spanning region and the cytosolic domain, completely.

This loss of so many domains in the putative protein variant would have important consequences on its performance. Among its described functions, KL has an important role in the maintenance of phosphorus homeostasis due to its participation in signaling pathways activated by FGF23, a major phosphatonin [23,24]. The presence of KL in a tissue is mandatory so that it is sensitive to the effects of FGF23. Actually, KL null mice develop all the phenotypes observed in Fgf23-/- mice [5]. This is explained by the fact that KL is a co-receptor of the FGF receptors [24]. When FGF23 binds to the FGFR1c-KL membrane complex, it triggers the intracellular mitogen-activated protein kinase (MAPK)/ERK signaling pathway which involves ERK1/2 phosphorylation and, eventually, upregulation of transcription regulators like the EGR1 gene [23,25]. The crystal structure of human 1:1:1 FGF23–FGFR1cecto–α-KLecto ternary complex at 3.0 Å resolution has been recently described [26]. According to this, both KL regions serves as a massive scaffold, tethering both FGFR1c and FGF23 to itself, enforcing FGF23-FGFR1c proximity and thus augmenting FGF23-FGFR1c binding affinity. Several residues in KL protein were described as essential for FGF23 signaling and most of them are missed in the putative p.Ile348Phefs*28 protein, like the receptor binding arm (RBA, residues 530-578); aminoacids D426, D745, C739 and D815 that bound Zn2+ ion serving as a prosthetic group in KL by minimizing interdomain flexibility and hence promoting co-receptor activity; and residues W417, K429, Y433, D756, I822 and I836 that participate in tethering of the flexible C-terminal tail of FGF23. Without all these residues, FGF23 signaling would be not possible since the interaction between FGF23, FGFR1c and KL would not take place."

 2) Experiments in transfected cell lines.

 Again, our apologizes to the reviewer.

 Regarding new experiments in transfected cell lines, it is necessary to take into account the special characteristics of the cells for these experiments. Specifically, these cells should express FGF23 receptors but not Klotho in order to check the FGF23-induced signaling pathways after transfecting plasmids with the KL full-length form, the p.Ile348Phefs*28 plasmid or plasmids with an empty vector. Based on these considerations, in our study we analyzed the activation of the FGF23-induced signaling pathways in the human cell line model HEK293. This is an optimal cell model since it endogenously expresses FGFR1c, the main receptor of FGF23, but does not express the Klotho protein (Turan and Ata, 2011; Diener et al.; 2015). This approximation allowed us determining the effects of carrying the p.Ile348Phefs*28 variant avoiding the interference of endogenous Klotho. HEK293 is a cell line widely employed for this kind of experimentation (Ichikawa et al. 2007; Wu et al. 2008; Farrow et al. 2009; Yamazaki et al. 2010; Lorenzi et al. 2010; Sakan et al. 2014; Ligumsky et al. 2015; Grabner et al. 2015; Diener et al. 2015; Chen et al. 2018, among others). We agree with the reviewer regarding the consideration that determining the absence of activation of the FGF23-induced signaling pathways in at least two different cell lines would be ideal. Nevertheless, to the best of our knowledge, we have not found other human cell lines that share these characteristics with HEK293 cells: absence of Klotho expression and presence of FGFR1c.

 All these considerations have been included in the Discusion section, and the lack of the study of FGF23-induced signaling pathways in other cell lines is now recognized as a limitation (lines 333-337 and 343-347). In addtion, two new references have been included in the manuscript (#27 and #28).

 References regarding the use of HEK293

·                     Turan, K.; Ata, P. Effects of intra- and extracellular factors on anti-aging klotho gene expression. Genet Mol Res. 2011,10(3), 2009-23. [doi: 10.4238/vol10-3gmr1261].

·                     Diener, S.; Schorpp, K.; Strom, T.M.; Hadian, K.; Lorenz-Depiereux, B. Development of A Cell-Based Assay to Identify Small Molecule Inhibitors of FGF23 Signaling. Assay Drug Dev Technol. 2015,13(8), 476-87. [doi: 10.1089/adt.2015.653].

·                     Ichikawa, S.; Imel, E.A.; Kreiter, M.L.; Yu, X.; Mackenzie, D.S.; Sorenson, A.H.; Goetz, R.; Mohammadi, M.; White, K.E.; Econs, M.J. A homozygous missense mutation in human KLOTHO causes severe tumoral calcinosis. J Clin Invest. 2007, 117(9), 2684-2691. [DOI: 10.1172/JCI31330]

·                     Wu, X.; Lemon, B.; Li, X.; Gupte, J.; Weiszmann, J.; Stevens, J.; Hawkins, N.; Shen, W.; Lindberg, R.; Chen, J.L.; Tian, H.; Li, Y. C-terminal tail of FGF19 determines its specificity toward Klotho co-receptors. J Biol Chem. 2008, 283(48), 33304-9. [doi: 10.1074/jbc.M803319200].

·                     Farrow, E.G.; Davis, S.I.; Summers, L.J.; White, K.E. Initial FGF23-mediated signaling occurs in the distal convoluted tubule. J Am Soc Nephrol. 2009, 20(5), 955-60. [doi: 10.1681/ASN.2008070783].

·                     Yamazaki, M.; Ozono, K.; Okada, T.; Tachikawa, K.; Kondou, H.; Ohata, Y.; Michigami, T. Both FGF23 and extracellular phosphate activate Raf/MEK/ERK pathway via FGF receptors in HEK293 cells. J Cell Biochem. 2010, 111(5), 1210-21. [doi: 10.1002/jcb.22842].

·                     Lorenzi, O.; Veyrat-Durebex, C.; Wollheim, C.B.; Villemin, P.; Rohner-Jeanrenaud, F.; Zanchi, A.; Vischer, U.M. Evidence against a direct role of klotho in insulin resistance. Pflugers Arch. 2010, 459(3), 465-73. [doi: 10.1007/s00424-009-0735-2].

·                     Sakan, H.; Nakatani, K.; Asai, O.; Imura, A.; Tanaka, T.; Yoshimoto, S.; Iwamoto, N.; Kurumatani, N.; Iwano, M.; Nabeshima, Y.; Konishi, N.; Saito, Y. Reduced renal α-Klotho expression in CKD patients and its effect on renal phosphate handling and vitamin D metabolism. PLoS One 2014, 9(1), e86301. [doi: 10.1371/journal.pone.0086301].

·                     Ligumsky, H.; Rubinek, T.; Merenbakh-Lamin, K.; Yeheskel, A.; Sertchook, R.; Shahmoon, S.; Aviel-Ronen, S.; Wolf, I. Tumor Suppressor Activity of Klotho in Breast Cancer Is Revealed by Structure-Function Analysis. Mol Cancer Res. 2015, 13(10), 1398-407. [doi: 10.1158/1541-7786.MCR-15-0141].

·                     Grabner, A.; Amaral, A.P.; Schramm, K.; Singh, S.; Sloan, A.; Yanucil, C.; Li, J.; Shehadeh, L.A.; Hare, J.M.; David, V.; Martin, A.; Fornoni, A.; Di Marco, G.S.; Kentrup, D.; Reuter, S.; Mayer, A.B.; Pavenstädt, H.; Stypmann, J.; Kuhn, C.; Hille, S.; Frey, N.; Leifheit-Nestler, M.; Richter, B.; Haffner, D.; Abraham, R.; Bange, J.; Sperl, B.; Ullrich, A.; Brand, M.; Wolf, M.; Faul, C. Activation of Cardiac Fibroblast Growth Factor Receptor 4 Causes Left Ventricular Hypertrophy. Cell Metab. 2015, 22(6), 1020-32. [doi: 10.1016/jNaNet.2015.09.002].

·                     Chen, G.; Liu, Y.; Goetz, R.; Fu, L.; Jayaraman, S.; Hu, M.C.; Moe, O.W.; Liang, G.; Li, X.; Mohammadi, M. α-Klotho is a non-enzymatic molecular scaffold for FGF23 hormone signalling. Nature 2018; 553(7689), 461-466.

J. Clin. Med. EISSN 2077-0383 Published by MDPI AG, Basel, Switzerland RSS E-Mail Table of Contents Alert
Back to Top